# Design, Synthesis, and Validation of a Novel [^11^C]Promethazine PET Probe for Imaging Abeta Using Autoradiography

**DOI:** 10.3390/molecules26082182

**Published:** 2021-04-10

**Authors:** Clayton A. Whitmore, Mariam I. Boules, William J. Behof, Justin R. Haynes, Dmitry Koktysh, Adam J. Rosenberg, Mohammed N. Tantawy, Wellington Pham

**Affiliations:** 1Vanderbilt University Medical Center, Vanderbilt University Institute of Imaging Science, Nashville, TN 37232, USA; clayton.whitmore@vumc.org (C.A.W.); mariam.i.boules@vumc.org (M.I.B.); william.j.behof@vumc.org (W.J.B.); j.r.haynes@vumc.org (J.R.H.); adam.j.rosenberg@vumc.org (A.J.R.); n.tantawy@vumc.org (M.N.T.); 2Department of Radiology and Radiological Sciences, Vanderbilt University Medical Center, Nashville, TN 37232, USA; 3Department of Chemistry, Vanderbilt University, VU Station, Nashville, TN 37235, USA; dmitry.koktysh@Vanderbilt.Edu; 4Vanderbilt Institute of Nanoscale Science and Engineering, Vanderbilt University, Nashville, TN 37235, USA; 5Department of Biomedical Engineering, Vanderbilt University, Nashville, TN 37235, USA; 6Vanderbilt Brain Institute, Vanderbilt University, Nashville, TN 37232, USA; 7Vanderbilt Ingram Cancer Center, Nashville, TN 37232, USA; 8Vanderbilt Institute of Chemical Biology, Vanderbilt University, Nashville, TN 37232, USA; 9Vanderbilt Memory and Alzheimer’s Center, Vanderbilt University Medical Center, Nashville, TN 37212, USA; 10Institute of Imaging Science, Vanderbilt University, 1161, 21st Avenue South, Nashville, TN 37232, USA

**Keywords:** promethazine, PET imaging, Abeta, Alzheimer, autoradiography

## Abstract

Promethazine, an antihistamine drug used in the clinical treatment of nausea, has been demonstrated the ability to bind Abeta in a transgenic mouse model of Alzheimer’s disease. However, so far, all of the studies were performed in vitro using extracted tissues. In this work, we report the design and synthesis of a novel [^11^C]promethazine PET radioligand for future in vivo studies. The [^11^C]promethazine was isolated by RP-HPLC with radiochemical purity >95% and molar activity of 48 TBq/mmol. The specificity of the probe was demonstrated using human hippocampal tissues via autoradiography.

## 1. Introduction

Alzheimer’s disease (AD) is the most common cause of dementia and is predicted to have a prevalence of 14 million in the U.S. by 2060, generating a burden that will become a public health crisis [1]. The mechanism that regulates neuronal degeneration in AD remains unknown. However, the definitive cytopathologic hallmarks of the disease include the formation and aggregation of amyloid-β proteins (Abeta) and intracellular hyperphosphorylated tau tangles, which together lead to profound neuron toxicity [2,3]. According to the latest diagnostic criteria, the clinical onset of AD spans over decades with three stages, including preclinical, mild cognitive impairment (MCI), and AD dementia [4,5,6,7]. According to the Abeta cascade hypothesis, the preclinical stage of AD-related pathology typically begins several decades before the onset of AD symptoms during the preclinical phase of AD [7,8]. This large time elapsed during these different stages of disease progression, particularly, during the preclinical stage opens up a window of opportunity for diagnosis.

The identification of amyloid-binding compounds is a crucial step in the development of imaging probes and therapeutics for the detection and treatment of Alzheimer’s disease [9,10], respectively. Unfortunately, the process typically lags during the translation from in vitro to in vivo studies due to the poor availability of the drugs in the brain [11,12]. Further, another limitation of probe development for AD is the lack of diversification of the chemical structures. Usually, new probes are deduced from near-identical structures of a known probe, and currently, there are two major families of Abeta probes being tested in clinical trials, including benzothiazole (Pittsburg compound B, Flutemetamol) backbone and stilbenes (Florbetapir, Florbetaben) structure. These drawbacks have necessitated the development of alternative methods to identify new Abeta-binding molecules to diversify the chemical structure. In parallel, this development will offer the potential to expand new binding motifs to enhance the chance for the hit, given Abeta possesses a wide diversity structure [13]. In the recent past, we witnessed a significant increase in the chemical development of novel Abeta probes utilizing a rational design approach [14,15,16,17].

In this work, we have undertaken an approach that combines high-throughput screening with MALDI imaging mass spectrometry (MALDI-IMS) to screen known compounds and repurpose their properties for Abeta-binding property. Through this work, we identified an antihistamine compound, promethazine, that can bind to amyloid plaques [18,19]. We also demonstrated that promethazine can cross the BBB, that it is retained more in the amyloid-burdened brain compared to a normal brain, and that its distribution within the brain corroborates with that of amyloid plaques [19]. To translate this molecule as a probe for future in vivo study, the main objective of this work focuses on the development of a [^11^C]promethazine ([^11^C]PMZ) PET radioligand. Furthermore, the proof-of-principle data were generated to demonstrate the future utility of the probe for in vivo applications. Toward that goal, we report herein the specificity of ([^11^C]PMZ radioligand as confirmed in human brain specimens.

## 2. Results

### 2.1. Design and Synthesis of a [^11^C]PMZ Precursor

This is the first report on the design and synthesis of a precursor for generating [^11^C]PMZ radioligand. Promethazine belongs to the phenothiazine family, and thus it is worthwhile to start the synthesis with phenothiazine **1** (Figure 1). Alkylation of the secondary amine with chloroacetone **2** in the presence of NaHCO_3_ in refluxing acetonitrile provided the ketone **3** with a decent yield. From compound **3**, PMZ **5** was formed through reductive amination using sodium triacetoxyborohydride and dimethylamine HCl salt in dimethylacetamide to provide a good yield. This compound is used as a standard “cold” compound for characterization of the [^11^C]labeling counterparts, as well as for blocking study to assess the specificity of the Abeta binding molecule. Using similar chemistry, albeit with methylamine HCl salt, resulted in the formation of the PET precursor **7**. It is worthwhile to mention the different reaction conditions used for obtaining compounds **5** and **7**. Since methylamine **6** is less bulky than dimethylamine **4**, the reaction with methylamine afforded a similar yield to that of dimethylamine while proceeding at a much lower reaction temperature.

### 2.2. [^11^C]CH_3_ Labeling Methods

For [^11^C] labeling, we explored three different methods to incorporate [^11^C]CH_3_ to precursor **7**, including loop-based synthesis with [^11^C]MeOTf, solution phase in the presence of sodium hydride with [^11^C]CH_3_I, and the solution labeling reactions with [^11^C]CH_3_I. We found the first and the last methods provided the best results, with the loop method giving the best molar activity (48 TBq/mmol), and the solution-phase giving the best radioactivity yield.

### 2.3. Specific Binding of [^11^C]PMZ Radioligand on Human Ex-Vivo Hippocampal Specimens

The ex vivo autoradiography of the hippocampi of human specimens (*n* = 12 slides from 2 patients) revealed a high-contrast topology of putative Abeta (Figure 2A,D,G). The images also show clusters of punctate labeling in the hippocampus. In contrast, very faint or no signals were detected in the consecutive slides pre-blocked with excess “cold” PMZ (Figure 2B,E) or anti-Abeta antibodies (4G8, BioLegend, San Diego, CA, USA) (Figure 2h). The signal intensity between [^11^C]PMZ-treated versus [^11^C]PMZ + PMZ-treated specimens shown in AB, DE, GH was quantified and presented in Figure 2C,F,I respectively.

### 2.4. Immunohistochemical Staining of Abeta in Human Ex Vivo Hippocampal Specimens

Human brain slides were subjected to immunohistochemical staining against Abeta using 4G8 antibodies. The data demonstrated that the green fluorescence label of Abeta is overexpressed outside of the nuclei (nuclear staining, blue) and mostly everywhere within the narrowly observed region of interest (Figure 2J,K). These Abeta-prone regions coincided with higher regional binding values for [^11^C]PMZ, as seen in the autoradiography data.

### 2.5. Discussion

We have developed the radiosynthesis of the novel [^11^C]PMZ radioligand for in vivo imaging applications. PMZ was first identified from our HTS assay designed to search for new Abeta-binding molecules [19]. As in the case of PMZ, we not only discovered new compounds [18], but the assay also enabled us to repurpose known drugs for a different application. Thus, our approach enhances the likelihood to improve Abeta inhibitors by offering unexplored binding motifs and new chemical genetics to advance AD research. Since PMZ belongs to a phenothiazine family, which has a large repertoire of many chemical structures, it is also suitable for structure-activity relationship (SAR) analysis to improve specificity with well-understood pharmacokinetics and safety profiles.

The chemistry shown in Figure 3 is applicable for the synthesis of PMZ as well as its modified analogs, in the same manner as we developed the precursor **7**. Since the phenothiazine backbone is very stable, elevated temperature can be used to increase the rate of the reaction and improve yield. For [^11^C]CH_3_ methylation, three different methods have been explored for the labeling of the PMZ precursor. The advantage of loop chemistry involves the use of a minute amount of samples and solvents, enabling the labeling operation in high concentration, facilitating methylation outcome [20,21]. Unfortunately, the yield proved to be sub-optimal in our labeling efforts. One of the issues we hypothesize is that loop chemistry is critically solvent-dependent [22] due to encountering small surface areas and the use of a remarkably small amount of solvent. Efforts are underway to optimize the production of [^11^C ]PMZ for both molar activity and radioactive yield.

Meanwhile, it is quite surprising that the addition of sodium hydride did not improve the reaction. Theoretically, the use of sodium hydride is ideal in this reaction since it is a common reagent to generate activated substrate in nucleophilic reactions. In this particular case, sodium hydride is used to deprotonate the amine for the promotion of its nucleophilic substitution. However, in contrast to our expectation, the reaction using sodium hydride was unproductive and resulted in poor yield. It is possible that sodium hydride not only served as a base but also a reducing agent. This dual ability, serving as a base and a reducing agent, in the presence of an electrophile like methyl iodide may result in byproducts when dimethylformamide was used in the reaction, as observed in the past [23].

Phenothiazine analogs have versatile applications across many diseases, including antimicrobial effects [24,25,26], antitumor effects [27,28,29,30,31,32,33], and potential roles in the treatment of neurodegenerative disease [34,35,36,37,38,39,40,41]. In the past, we showed PMZ binds to Abeta available in the brain lysate of 5XFAD mice in an in vitro study. We also demonstrated that intravenously injected PMZ resulted in the molecules being distributed to the brain and that PMZ retention in the brain is Abeta dependent [19]. In this work, we further confirm the specificity of the PMZ using human specimens. Autoradiography data on postmortem human brain sections using [^11^C]PMZ showed that binding was observed throughout the hippocampal region (please note our device could image only a small FOV, 1-inch × 1.25 inch). When consecutive slides were treated with either “cold” PMZ or anti-Abeta antibodies, the binding reduces significantly. Significant differences in both human samples were observed in a comparative manner (7- to 8-fold, *n* = 4). The same holds true when the tissues that were treated with anti-Abeta antibodies before exposure to [^11^C]PMZ. We also observed similar binding characters using the hippocampal slices obtained from 5XFAD mice (Appendix A in Appendix A). There are some regions of binding in the antibody-treated tissue. This is probably caused by an epitope-binding mismatch between PMZ and antibodies.

## 3. Materials and Methods

### 3.1. Synthesis of a Promethazine Precursor for [^11^C] Labeling

*1-(10H-phenothiazin-10-yl)propan-2-one***3**. To a stirring solution containing phenothiazine 1 (2.5 g, 12.5 mmol), NaI (3.7 g, 25 mmol), and NaHCO_3_ (2.1 g, 25 mmol) in acetonitrile (50 mL), chloroacetone 2 (1.27 g, 13.75 mmol) was added and heated to 80 °C over the course of 48 h (Figure 1). An additional 1 mL of 2 was added and heated overnight. The reaction was cooled, diluted with H_2_O and EtOAc. The product was extracted with EtOAc (3×). The organic layers were combined, washed with saturated sodium thiosulfate, brine, dried over Na_2_SO_4_, filtered, and concentrated under reduced pressure. The crude product was purified by flash chromatography (0–50% EtOAc/Hexane over the course of 20 min, Teledyne ISCO HPLC Combiflash).

^1^H-NMR (400 MHz, DMSO-*d*_6_): 7.09 (m,4H); 6.91 (td, *J*1 = 8.0 Hz, *J*2 = 1.2 Hz, 2H); 6.53 (dd, *J*1 = 8.4 Hz, *J*2 = 0.8 Hz, 2H); 4.78 (s, 2H); 2.31 (s, 3H). ^13^C-NMR (400 MHz, DMSO-*d*_6_): 204.4, 143.8, 127.4, 126.5, 122.6, 121.2, 115.3, 57.4, 27.0.

*N,N-dimethyl-1-(10H-phenothiazin-10-yl)propan-2-amine***5**. To a stirring solution of ketone 3 (264 mg, 1.03 mmol) and dimethylamine HCl (1.1g, 14.3 mmol) in DMA (4 mL), sodium triacetoxyborohydride (880 mg, 5.15 mmol) was added and heated to 50 °C for 5 days. The reaction was cooled, diluted with 50 mL of 1 M NaOH and EtOAc. The product was extracted with EtOAc (3×). The organic layers were combined, washed with H_2_O-brine (3×), dried over Na_2_SO_4_, filtered, and concentrated under reduced pressure. Crude product was purified by flash chromatography (0–50% (80% CH_2_Cl_2_/18% MeOH/2% NH_4_OH)/CH_2_Cl_2_). Purified oil was redissolved in approximately 0.5 mL of 4M HCl in dioxane. The reaction mixture was let to stay at rt for 30 min and concentrated under reduced pressure to afford product 5.

^1^H-NMR (400 MHz, CDCl_3_): 7.23 (m, 4H); 7.07 (d, *J* = 7.8 Hz, 2H); 7.01 (m, 2H); 4.74 (dd, *J*1 = 14.0 Hz, *J*2 = 5.7 Hz, 1H); 3.91 (dd, *J*1 = 14.0 Hz, *J*2 = 7.2 Hz, 1H) 3.72 (m, 1H); 2.79 (d, *J* = 5.0 Hz, 3H), 2.75 (d, *J* = 5.0 Hz, 3H); 1.51 (d, *J* = 6.7 Hz, 3H). ^13^C-NMR (400 MHz, CDCl_3_): 144.3, 128.2, 128.1, 126.7, 123.9, 116.2, 58.3, 48.2, 40.2, 39.4, 12.8.

*N-methyl-1-(10H-phenothiazin-10-yl)propan-2-amine***7**. To a stirring solution of ketone 3 (500 mg, 1.96 mmol) and methylamine HCl 6 (1.0 g, 15.7 mmol) in DMA (8 mL), sodium triacetoxyborohydride (2 g, 9.8 mmol) was added and stirred overnight. The reaction was cooled, diluted with 50 mL of 1 M NaOH and EtOAc. The product was extracted 3× with EtOAc. The organic layers were combined, washed with H_2_O-brine (3×), dried over Na_2_SO_4_, filtered, and concentrated under reduced pressure. The crude product was purified by flash chromatography (0–50% (80% CH_2_Cl_2_/18% MeOH/2% NH_4_OH)/CH_2_Cl_2_) to afford the precursor 7 (Figure 1).

^1^H-NMR (400 MHz, CDCl_3_): 7.16 (m, 4H); 6.93 (m, 4H); 3.84 (m, 2H); 3.05 (m, 1H); 2.36 (s, 3H); 1.13 (d, *J* = 6.3 Hz, 3H). ^13^C-NMR (400 MHz, CDCl_3_): 145.7, 127.8, 127.4, 126.3, 122.9, 116.2, 53.5, 51.6, 34.0, 17.9.

### 3.2. Labeling the Promethazine Precursor with [^11^C]CH_3_ Radioisotope and Purification

[^11^C]PMZ was prepared using the GE Healthcare Tracerlab FXC-Pro, a commercially supplied reaction platform. The [^11^C]CO_2_ is made by irradiating a target filled with nitrogen and 1% oxygen gas with protons. The [^11^C]CO_2_ is then trapped on nickel Shimalite with molecular sieves at room temperature. The [^11^C]CO_2_ is then converted to [^11^C]CH_4_ by heating the trapped [^11^C]CO_2_ to 400 °C in the presence of hydrogen gas. The [^11^C]CH_4_ is then released from the nickel Shimalite at 400 °C and isolated on molecular sieves at −75 °C. The [^11^C]CH_4_ is then converted to [^11^C]MeI via a recirculation through gaseous iodine at ~720 °C, with the [^11^C]MeI being trapped on Porapak N with each cycle. The [^11^C]MeI (typically 500–700 mCi) is then released from the Porapak N by heating with a gentle flow of helium (10–15 mL/min) that is either directed directly to the reaction mixture or passed through an AgOTf impregnated column at ~200 °C to convert the [^11^C]MeI to [^11^C]MeOTf; this [^11^C]MeOTf is then transferred to the reaction mixture with a gentle flow of helium (10–15 mL/min).

#### 3.2.1. Loop Method

The [^11^C]MeOTf is transferred to a 5 mL stainless-steel loop containing a mixture of the Promethazine Precursor (1.0 mg) and 2-Butanone (200 μL) at room temperature. Once all radioactivity has been eluted from the Porapak N, the reaction mixture is allowed to react at room temperature for 1 min, after which time the loop is flushed with the mobile purification phase (50% Acetonitrile in 100 mM ammonium formate) directly onto the purification column (Phenomenex Luna C18(2), 250 mm × 10 mm, 10 μm, Aschaffenburg, Germany). The desired radioactive peak is isolated and diluted with water (20 mL) followed by transfer on a C18 Sep-Pak Plus. The Sep-Pak is then washed with water (10 mL) and subsequently eluted with ethanol (1 mL) followed by 0.9% saline (10 mL). This mixture is then passed through a 0.22 μm sterilizing filter and into the final vial.

#### 3.2.2. Solution Method

[^11^C]MeI is bubbled into a reactor containing a mixture of the promethazine precursor (1.0 mg), sodium hydride (1.7 mg, if used), and DMF (200 μL) at ~0 °C. Once all radioactivity has been eluted from the Porapak N, the reaction mixture is heated to 90 °C for 2 min, cooled to room temperature, diluted with 1.7 mL of the mobile purification phase (50% Acetonitrile in 100 mM ammonium formate), and injected onto the purification column (Phenomenex Luna C18(2), 250 mm × 10 mm, 10 μm). The desired radioactive peak is isolated and diluted with water (20 mL) followed by transfer on a C18 Sep-Pak Plus. The Sep-Pak is then washed with water (10 mL) and subsequently eluted with ethanol (1 mL) followed by 0.9% saline (10 mL). This mixture is then passed through a 0.22 μm sterilizing filter and into the final vial.

The radiochemical purity and the identity of the ([^11^C]PMZ radioligand were characterized using an analytical HPLC system, equipped with a UV absorption detector (λ = 254 nm) and a radio-detector (Bioscan Flow-Count, Eckert & Ziegler, Wilmington, MA, USA) (Figure 3C). The chromatography setup, including Phenomenex Luna C18(2) (250 mm × 4.6 mm, 5 μm) with a typical mobile phase of acetonitrile and aqueous ammonium formate (40/60). The identity of the [^11^C]PMZ was confirmed by comparing the retention time (approximately 8.8 min) with co-injected and unlabeled promethazine **5** along with the gamma peak (Figure 3C).

The molar activity of [^11^C]PMZ was assessed using the same HPLC system and conditions as described above. The molar activity was approximately 48 TBq/mmol for the loop method and approximately 8 TBq/mmol for the solution phase.

### 3.3. Autoradiography

Brain tissue. Human brain samples (*n* = 2 donors) were provided by Harvard Brain Tissue Resource Center in dry ice and were immediately stored at −80 °C. The frozen hippocampal samples (4 cm × 4 cm) were cryosectioned at 8–10 μm, transferred to microscope glass slides, and kept at −80 °C before use. Two consecutive and nearly identical samples were placed on each slide before being treated with [^11^C]PMZ or [^11^C]PMZ + PMZ for direct comparison. The specific activity of [^11^C]PMZ was approximately 0.08 TBq/mg.

Before autoradiography, one specimen from each slide was treated with excess “cold” promethazine (100 μM) or anti-Abeta antibodies (4G8) for 10–15 min by dipping the slide into either solution so as to fully submerge only one (the bottom) sample. The samples were washed with PBS twice and dried using a small stream of warm air prior to dipping the whole glass slide into an incubation chamber containing a solution with a concentration of approximately 3.23–5 mCi of ([^11^C]PMZ radioligand in 500 mL of PBS buffer. After 20 min of incubation at room temperature, the samples were rinsed twice in PBS for 1 min each, followed by rapid exposure to distilled water. Slides were then placed on Kimwipe papers under a gentle stream of warm airflow to assist with drying. The glass slides were exposed to the Beta Imager (Bio Space Lab, France) in the way that the field-of-view (FOV) would cover two tissues to enable imaging simultaneously for a duration of 60 min. After that, the regions-of-interest (ROIs) were drawn around each section in the autoradiography images, and the decay-corrected counts per minute (CPM) with each ROI were compared between sections.

### 3.4. Abeta Immunohistochemistry

Immunohistochemistry (IHC) imaging was done on male and female human hippocampus sections. The sections were embedded in OCT (optimal cutting temperature), cryo-sectioned (10 µm), and stained for the presence of Abeta plaques. Sections were allowed to dry, followed by being washed in PBS twice for 5 min and once for 10 min. After washing, the sections were blocked in blocking buffer (5% normal goat serum [NGS], 10% Triton X-100, and 0.5% bovine albumin in PBS) for 1 h at room temperature. Slides were then incubated overnight at 4 °C with Abeta antibody (1.5 µg/mL in blocking buffer, 4G8). The following day, the sections were washed 3 times for 10 min in PBS. Slides were then incubated for one hour at room temperature with Alexa Fluor 488 secondary antibody (1 µg/mL in blocking buffer). After incubation, the secondary antibody was removed, and the sections were rinsed in PBS twice for 10 min and a third time for 30 min. Then, the slides were washed briefly and mounted with a mounting medium with DAPI, H-1200 (Vector Laboratories, San Francisco, CA, USA). A coverslip was placed on top of the sections and sealed with nail polish. Two sections of both male and female hippocampal regions were imaged using a Zeiss fluorescence microscope (Axio Observer, Carl Zeiss, Jena, Germany). The DAPI channel was used with an excitation of 353 nm and an emission of 465 nm; the Alexa Fluor 489 dye was excited at 493 nm, with an emission at 517 nm. All the shown data used a 40× magnification lens with an exposure time of 150 ms.

### 3.5. Data Processing and Quantification

All autoradiography data were converted to an 8-bit format. Then, the background was universally adjusted to eliminate unspecified signals, followed by thresholding using the Otsu method and conversion of the signal into binary information before pixel quantification using ImageJ. The difference between means with the standard error in the [^11^C]PMZ vs [^11^C]PMZ + PMZ groups was evaluated using an unpaired *t*-test.

### 3.6. Statistical Analysis

The experimental data were evaluated with a standard unpaired *t*-test between tissues treated with [^11^C]PMZ and ^11^C]PMZ + PMZ using GraphPad software (La Jolla, CA, USA). *p* values are two-tailed, and differences with *p* values < 0.05 were considered statistically significant.

## 4. Conclusions

In conclusion, we report the design, synthesis, and evaluation of a novel [^11^C]PMZ probe for potential in vivo imaging of Abeta. The advantage of using PMZ as a backbone for developing Abeta-binding molecules is its ability to cross the BBB. Further, promethazine belongs to a large family of phenothiazines, thus offering a repertoire of candidates with diverse chemical structures, facilitating future modifications to improve either binding specificities, pharmacokinetics, or safety profiles.

## Figures and Tables

**Figure 1 molecules-26-02182-f001:**
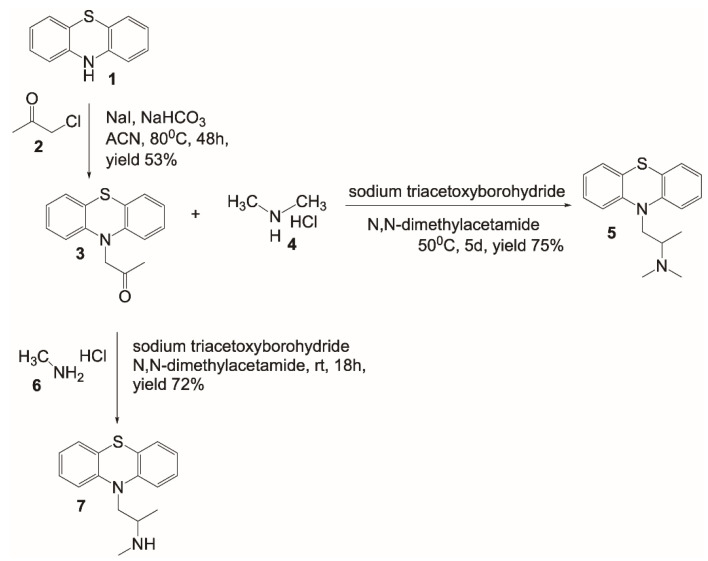
Design and synthesis of PMZ and a precursor for the synthesis of [^11^C]PMZ radioligand.

**Figure 2 molecules-26-02182-f002:**
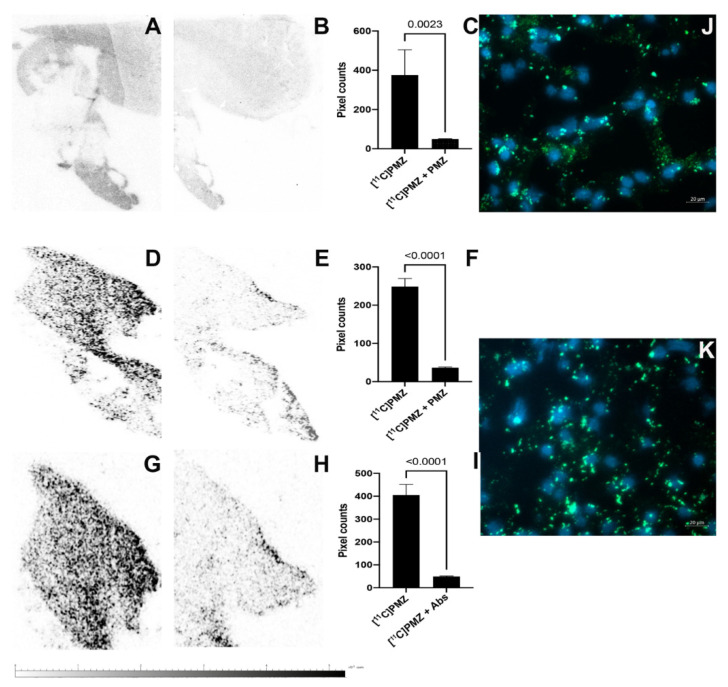
Ex vivo autoradiography data from two human hippocampal tissues using [^11^C]PMZ radiotracers (Female specimen (**A**–**C**,**J**) and male specimen (**D**–**I**,**K**)). Representative autoradiograms of [^11^C]PMZ binding in the tissues (**A**,**D**,**G**); (**B**,**E**) [^11^C]PMZ binding in the presence of an excess of PMZ or (**H**) 4G8 antibodies; (**C**,**F**,**I**) quantification of binding of [^11^C]PMZ for (**A**/**B**,**D**/**E**,**G**/**H**, *n* = 4 slides, each), respectively. Significant differences were observed in [^11^C]PMZ-treated tissues versus the blocking counterparts (the difference between **A**&**B**,**D**&**E**,**G**&**H** are 8-, 7-, and 8-fold with *p* = 0.0023, *p* < 0.0001, and *p* < 0.0001, respectively). (**J**,**K**) Immunohistochemical detection of Abeta, green (488 nm) immunoreactivity depicts Abeta; blue (405 nm) depicts DAPI labeling for nuclei.

**Figure 3 molecules-26-02182-f003:**
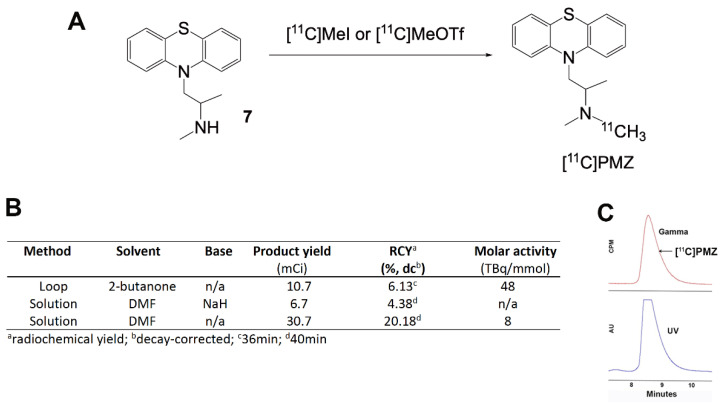
(**A**) General scheme for the synthesis of [^11^C]PMZ PET radioligand via different methods of methylation; (**B**) different methods of [^11^C]CH_3_ labeling along with reaction conditions and yields; (**C**) Analytical HPLC chromatogram of [^11^C]PMZ with co-injection of the cold standard. Bottom, ultraviolet (UV), absorbance at 254 nm. Top, gamma-detection of radioactivity. The retention time for [^11^C]PMZ at the described HPLC condition is approximately 8.8 min.

## Data Availability

Data is contained within the article or Appendix A.

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
