# Peer review of "Design, Synthesis, and Validation of a Novel [^11^C]Promethazine PET Probe for Imaging Abeta Using Autoradiography"

_molecules, 2021, doi:10.3390/molecules26082182_

Round 1

Reviewer 1 Report

Overall Comments:

-The entire document is full of grammar errors that need to be addressed. A thorough proof reading should be done to correct the error. To many to list.

- The design of the studies appears to be sound, but the presentation of the results needs to be cleaned up to be an accurate representation of those results

Specific Comments by line number (grammatical errors are not listed):

  • 35: The "10 million by 2050" comment needs to have a location specified. Is this worldwide or for a specific country/region?
  • 47: The term "cure" should be changed to "treatment". 
  • 49: "Impenetrable" is not an appropriate descriptor. There are plenty of drugs that penetrate the BBB.
  • 52: The statement that drugs are developed from a "known probe" implies there is one drug. The authors then contradict themselves by saying there are multiple drug motifs. Reword to be clear
  • 86-88: Reductive aminations using primary versus secondary amines and the reaction rate for each process are well understood. This reviewer feels that this statement is not a unique or valuable observation.
  • Figure 1: Bond lengths and bond angles need to be cleaned up. The scheme does not look polished. 
  • 96: The statement of molar activity does not fit in the prior sentence. Is this the best molar activity, average, or specific to a process?
  • 97: The statement "quick reaction time" does not make sense considering that the loop method has been used, which surely has a shorter reaction time. 
  • Section 2.3 Overall: This section is difficult to follow. The entirety of the section should be reworded to flow better and hopefully explain the steps taken in the process
  • 104: This is the first time that "anti-Abeta antibodies" have been mentioned. The identity of the antibodies should be provided at this point. 
  • Figure 3 and Figure 2 are out of order. 
  • The current Figure 2 needs a log of work: 1) Part A is missing the C-11 in the methyl iodide. The authors also are clear that they used methyl triflate for the preparation, this figure should reflect that option. The degree symbol is not correct, it's clearly a super scripted zero. 2) Part B is not accurate table for the results discussed later. This should reflect what was actually done, including the use of methyl triflate. 3) Part C. If this is representative example of a prep-HPLC, there is no way any the molar activity was 48 TBq/mmol. The UV is maxed out. Something is wrong with the supplied spectrum
  • 154-156: This directly contradicts what the authors report in section 2.2. 
  • 160: "sodium hydride not only served as a base but also as a source of hydride." The authors should clearly state that NaH is potentially serving as a base and a reducing agent. It goes without saying that sodium hydride is acting as a base and a source of hydride. The authors should also give a little more detail on the proposed impurities from the DMF decomposition and if they have any proof of the formation.
  • 190: More detail needs to be provided for the flash chromatography conditions. 0-50% EtOAc/Hexane over what timeframe? What system is being used for the flash chromatography?
  • 198: subscripts for sodium sulfate are missing. 
  • NMR Data: extensive formatting needs to be done for this information. All coupling constants should be reported in Hz. In many cases, spacing and subscripts missing. 
  • 229: This is the first mention of methyl triflate. Should be discussed earlier in the transcript.
  • 232: What size loop is being used for this process and what is the flow rate of the methyl triflate for this step?
  • 257-259: This section now says that Figure 2C is of an analytical co-injection trace. Which directly contradicts the Figure.
  • 260: No data from the LC-MS/MS is presented. Furthermore, it's implied that the LC-MS/MS was done on the C-11 compound. I find this hard to believe. This needs to be clarified. 
  • 261: What does "specific radioactivity" mean? Is this supposed to be molar activity?
  • 262: The Molar activity should be provided in a range with an "n = ?" value. 
  • 285: "OCT" needs to be defined. 

Author Response

We appreciate the reviewers for their compliments about the work, as well as for taking their precious time to perform the review and provide suggestions to improve the quality of our manuscript. We answered all of the major and minor issues raised by the reviewers in this Introduction. Further, along with the response herein, we also updated the revised manuscript with the appropriate modifications as suggested by the reviewers. In this resubmission, we incorporate a version with all of the updated parts labeled in red fonts for review-purpose only, and the other identical and clean version.

Reviewer 1

  1. The entire document is full of grammar errors that need to be addressed. Thorough proofreading should be done to correct the error. Too many to list.

Answer: We have already corrected this issue in the revised manuscript.

  1. The design of the studies appears to be sound, but the presentation of the results needs to be cleaned up to be an accurate representation of those results.

Answer: We agreed with the reviewer regarding this problem. We have already made a significant reshuffle of the figures; the Results and Materials & Methods description match with the data shown in the figures. In addition, we also discussed the results based on the orders listed in now Figure 3C. Overall, all of the confusion has been alleviated in this revised version.

  1. The "10 million by 2050" comment needs to have a location specified. Is this worldwide or for a specific country/region?

Answer: We are sorry for not making the statistics clearer in the last submission. We updated the information in the Introduction.

  1. The term "cure" should be changed to "treatment"

Answer:  We updated this new term as suggested by the reviewer.

  1. "Impenetrable" is not an appropriate descriptor. There are plenty of drugs that penetrate the BBB.

Answer: This expression has already been rephrased appropriately in the revised manuscript. And this can be found in the Introduction.

  1. The statement that drugs are developed from a "known probe" implies there is one drug. The authors then contradict themselves by saying there are multiple drug motifs. Reword to be clear

Answer: We kindly disagree with this idea, as we showed the evidence that many new Abeta probes were developed based on the structures of a known probe. For instance, flutemetamol was developed several years after Pittsburg compound B. The only difference is that the former is the 18F-version where the isotope was labeled on the aromatic ring, while later was labeled as a [11C]CH3 labeled product.

  1. Reductive aminations using primary versus secondary amines and the reaction rate for each process are well understood. This reviewer feels that this statement is not a unique or valuable observation.

Answer: We agreed with the reviewer about this notion. However, this paper targets a wider group of readers of different backgrounds, and thus we feel that a detailed discussion of the experimental results may be helpful to those who do not have a strong chemistry background.

  1. Bond lengths and bond angles need to be cleaned up. The scheme does not look polished. 

Answer: We thank the reviewer for pointing out this issue. Figure 1 has been modified appropriately.

  1. The statement "quick reaction time" does not make sense considering that the loop method has been used, which surely has a shorter reaction time. 

Answer: We agreed with the reviewer about this statement. It has been removed from the revised manuscript.

  1. Section 2.3 Overall: This section is difficult to follow. The entirety of the section should be reworded to flow better and hopefully explain the steps taken in the process

Answer: We thank the reviewer for pointing out this issue. We also accept our oversight when preparing this section. It has now been fixed in the revised manuscript.

  1. This is the first time that "anti-Abeta antibodies" have been mentioned. The identity of the antibodies should be provided at this point. 

Answer: We updated this information in the revised version.

  1. Figure 3 and Figure 2 are out of order. 

Answer: We switched the order of these 2 figures in the revised manuscript.

  1. The current Figure 2 needs a lot of work: 1) Part A is missing the C-11 in the methyl iodide. The authors also are clear that they used methyl triflate for the preparation, this figure should reflect that option. The degree symbol is not correct, it's clearly a super scripted zero. 2) Part B is not accurate table for the results discussed later. This should reflect what was actually done, including the use of methyl triflate. 3) Part C. If this is representative example of a prep-HPLC, there is no way any the molar activity was 48 TBq/mmol. The UV is maxed out. Something is wrong with the supplied spectrum

Answer: Figure 3 (in the revised version) has been corrected to reflect the proper labeling condition as discussed in the Results. Further, we also discussed the three methods of [11C]CH3 in the order that shown in now Figure 3B in this revised version.

  1. Inconsistent statements about sodium hydride in Discussion against what have been told in the Results (section 2.2 ).

Answer: We are sorry for the inconsistent preparation of the manuscript. Indeed, the discussion contradicts what has been shown in the now Figure 3B and the Results. We have modified the work in the related sections to ensure consistency and mitigated the above-mentioned mistakes.

  1. sodium hydride not only served as a base but also as a source of hydride." The authors should clearly state that NaH is potentially serving as a base and a reducing agent. It goes without saying that sodium hydride is acting as a base and a source of hydride. The authors should also give a little more detail on the proposed impurities from the DMF decomposition and if they have any proof of the formation.

Answer: We clarify NaH served as a base and a reducing agent as recommended. We did not isolate or characterize the impurities from DMF-based side products. But we listed a reference (J. Org. Chem. 2009, 74, 2567-2570), which showed that hydride reduction of DMF could result in the formation of sodium (dimethylamino)methanolate, then dimethylamine. Both of these nucleophiles could be the source that facilitates the side reactions and lower the yield of the desired product.

  1. More detail needs to be provided for the flash chromatography conditions. 0-50% EtOAc/Hexane over what timeframe? What system is being used for the flash chromatography?

Answer: We updated this data in the Materials & Methods section.

  1. Subscripts for sodium sulfate are missing.

Answer: We thank the reviewer for this reminder on the nomenclature of chemical formula; it is now fixed.

  1. NMR Data: extensive formatting needs to be done for this information. All coupling constants should be reported in Hz. In many cases, spacing and subscripts missing. 

Answer: Every issue raised in this area, has been modified appropriately.

  1. This is the first mention of methyl triflate. Should be discussed earlier in the transcript (from lines 229-Materials and Methods).

Answer: Based on this recommendation we started to discuss radiolabeling using methyl triflate in the legend of now Figure 3 (Discussion).

  1. What size loop is being used for this process and what is the flow rate of the methyl triflate for this step?

Answer: We use a 5mL stainless-steel loop. This data is now updated in the Materials & Methods.

  1. This section now says that Figure 2C is of an analytical co-injection trace. Which directly contradicts the Figure.

Answer: We are sorry for the confusion. We confirm that the data from now Figure 3C are analytical HPLC with co-injection of “cold” compound. And this data has been updated in the legend of Figure 3, to be consistent with the text shown in the Materials & Methods.

  1. No data from the LC-MS/MS is presented. Furthermore, it's implied that the LC-MS/MS was done on the C-11 compound. I find this hard to believe. This needs to be clarified. 

Answer: We are sorry for this significant oversight and lack of strenuous proof-reading of the manuscript. It is our mistake not communicating with the chemistry team carefully. There is no LC-MS/MS for the labeled compound.

  1. What does "specific radioactivity" mean? Is this supposed to be molar activity?

Answer: We corrected it, as molar activity.

  1. "OCT" needs to be defined.

Answer: We define this abbreviation in the revised manuscript.

Reviewer 2 Report

The authors describe the radiosynthesis and use of [11C]Promethazine as a PET probe for imaging amyloid-Beta proteins.

Overall the paper is well written but some issues must be fixed before publication.

In the paper and Abstract, the authors mentioned that only ex vivo/in vitro studies were performed so far , however in this work the same disadvantage could be pointed out as no in vivo results are described. Therefore a partial modification of the abstract and conclusion should be done (ie: potential in vivo imaging).

Line 78-80 Synthesis of cpd 3 si performed with NaHCO3 (according to scheme 1 and material and methods), NaHCO3 is not a Strong base as stated. Temperature is described as an "elevated temperature" while the temperature is only 80°C for a solvent having a boiling point of 81°C.

"From compound 3, PMZ 5 can be developed by treating dimethylamine HCl salt with a mild reducing reagent such as sodium triacetoxyborohydride" Please rephrase : dimethylamine is not treated by sodium triacetoxyborohydride, its the resulting imine/iminium salt , moreover "such as" is not necessary as only sodium triacetoxyborohydride was used in this work.

"For [11C] labeling, we explored three different methods to incorporate [11C]CH3 to precursor 7, including loop-based synthesis, the solution labeling, and the solution phase with the presence of a strong base, such as sodium hydride. We found the first and the last methods are reproducible, albeit the last method confers the best reaction yield, the molar activity of 48 TBq/mmol at the end of synthesis. This method is more robust than the other operations with a quick reaction time, suitable for short half-life isotopes." : This is a very confusing way to describe the process specially considering the results reported in figure 2B. if last method as mentioned is the one using NaH (not such as NaH ! except if the authors used other strong base not described here ?) then it has the lowest yield.

What was finally the method chosen by the authors, the last part of the sentence indicating a robust method refers to what ? according to figure2B results the first and second method (as depicted in the text = first and last in figure 2B) are giving the best yield. Methods should be clearly defined and set in the same order in the text as in the table 2B to avoid confusion.

2.3 Specific Binding

Promethazine is a known drug with a high affinity for H1 receptors, some of then being located in brain areas, was this consider ? could it be a drawback ? the point should be mentioned and some how elucidated.

Figure 2 and 3 are inverted (wrong numbering, Figure 2 should appear before figure 3)

2.5 Discussion

Line 125 : this is not a novel chemistry (N-methylation using 11C is well known) , should be rephrased as "We have developed a synthesis of ......."

Figure 2

The chemical reaction involving CH3I (should be written [11C]CH3I ) is different from the data provided in the material and methods (100°C, 5 min on scheme and 90°C, 2 minutes in the material and method ?)

2B: as mentioned above, table should be in same order as the text.

2B: the author indicate a specific activity at the end of synthesis of 48 TBq/mmol. the value could be added in the table, specially if a specific activity was also determined for the two other methods.

2C: is described in the bottom note as a preparative HPLC, while in material and methods it refers to an analytical HPLC co-injected with "cold" reference (line 253-260)

Could you also please explain why in the preparative HPLC the UV signal reach a saturation (considering the low amount of precursor used this seems surprising).

Why did you choose NaH for a methylation reaction on a secondary amine, this is a very uncommon way to proceed, only weak base are necessary and i doubt of the deprotonation of the secondary amine by NaH.

Line 154-156: Author state that use of NaH is ideal ....it is a common reagent to generate activated substrate " this is true only considering the acidity of the substrate (ie alcohols, thiols). Most 11C methylation of amines  are carried out with weak base (NaOH, Na2CO3, CsCO3..). Moreover the failure of the reaction with NaH and the success without the base demonstrate it. Why a reaction with a weak base was not attempted to compare NaH with an other more suitable base (used only to trap HI) ?

I am not convince by the postulated action of NaH proposed by the authors and unless they can add a relevant reference i would recommend not to discuss that issue.

line 163: Again this sentence is very confusing: "Despite all these drawbacks...as a solvent". To what are you referring ? the drawbacks are connected to a specific method (not commonly used, NaH) not to the general methylation process of secondary amines.

No radiochemical yields or conversion are given, this is surprising, only product yield in mCi (official unit is Bq or MBq) is indicated in table 2B without knowing the initial activity introduced in the synthesis and if the results are based on [11C]CH3I or [11C]CH3OTF. 

The value should be indicated decay corrected or non decay corrected along with the total synthesis time (purification and formulation included).

In material and methods: the process of methylation used by the authors should be clearly defined: general procedure are described referring to CH3I and CH3OTf while in the text the use of CH3OTf is not mentioned. What methylation agent was fused for each method ?

3.2.2 the solution method: this is a general procedure (base if used) with some difference between text and figure 2 (reaction time, temperature) and also between HPLC description (analytical) and figure 2C (preparative).

The method involving NaH should be clearly described as not mixed with the solution method selected by the authors.

a specific activity of 48 TBq/mmol. was obtained , is it a single determination or an average value (n= ?) 

Line 259 "further the ...using LC-MS/MS. this sentence is useless here and should be in the chemical characterization of the reference compound (with the observed value for MS peak) as for 11C-PMZ the MS value observed correspond only to the reference compound.

in material and methods: Phenomenex C18(2) , Phenomenex is a brand not a model of column, please indicate the name of the column used, to what refers the (2) ? some more detail on HPLC would be welcome (HPLC model, injection loop volume, flow) 

Retention time of 11C-PMZ should be indicated clearly in the material and methods section.

Line 198 : Na2SO4 (Typo)

Line 200-201: approximtely = approximately (typo)

line 201: dioxanes (why plurial ?)

For Autoradiography: What was the specific activity of 11C-PMZ at the time of incubation ?

Analytical HPLC chromatograms should be included in the supplementary materials, with and without co-injection of "cold" reference compound.

the preparative chromatogram of a typical 11C-PMZ radiosynthesis should also be added.

In the references list, check the abbreviation, i guess a dot should be added on abbreviated terms (Chem. Commun. instead of Chem Commun)

The coma after the year does not need to be in bold.

Author Response

We appreciate the reviewers for their compliments about the work, as well as for taking their precious time to perform the review and provide suggestions to improve the quality of our manuscript. We answered all of the major and minor issues raised by the reviewers in this Introduction. Further, along with the response herein, we also updated the revised manuscript with the appropriate modifications as suggested by the reviewers. In this resubmission, we incorporate a version with all of the updated parts labeled in red fonts for review-purpose only, and the other identical and clean version.

  1. In the paper and Abstract, the authors mentioned that only ex vivo/in vitro studies were performed so far, however, in this work the same disadvantage could be pointed out as no in vivo results are described. Therefore, a partial modification of the abstract and conclusion should be done (ie: potential in vivo imaging).

Answer: In the Abstract, we mentioned the use of the probe for “future in vivo studies” and we added “for potential in vivo imaging of Abeta” in the Conclusion.

  1. Synthesis of compound 3 is performed with NaHCO3 (according to scheme 1 and material and methods), NaHCO3 is not a Strong base as stated. Temperature is described as an "elevated temperature" while the temperature is only 80°C for a solvent having a boiling point of 81°C.

Answer: We thank the reviewer for reminding this error; we have updated the information appropriately, and this updated information can be found in the Results section.

  1. “From compound 3, PMZ 5 can be developed by treating dimethylamine HCl salt with a mild reducing reagent such as sodium triacetoxyborohydride" Please rephrase : dimethylamine is not treated by sodium triacetoxyborohydride, its the resulting imine/iminium salt , moreover "such as" is not necessary as only sodium triacetoxyborohydride was used in this work.

Answer: We rephrased this sentence in the revised manuscript.

  1. For [11C] labeling, we explored three different methods to incorporate [11C]CH3 to precursor 7, including loop-based synthesis, the solution labeling, and the solution phase with the presence of a strong base, such as sodium hydride. We found the first and the last methods are reproducible, albeit the last method confers the best reaction yield, the molar activity of 48 TBq/mmol at the end of synthesis. This method is more robust than the other operations with a quick reaction time, suitable for short half-life isotopes." : This is a very confusing way to describe the process specially considering the results reported in figure 2B. if last method as mentioned is the one using NaH (not such as NaH ! except if the authors used other strong base not described here ?) then it has the lowest yield.

Answer: We thank the reviewer for the critical comment; the mismatch had resulted from a last-minute modification of the figure. We have modified this [11C]CH3 labeling section appropriately, and it is consistent with the order shown in now Figure 3B.

  1. What was finally the method chosen by the authors, the last part of the sentence indicating a robust method refers to what? according to figure2B results the first and second method (as depicted in the text = first and last in figure 2B) are giving the best yield. Methods should be clearly defined and set in the same order in the text as in the table 2B to avoid confusion.

Answer: We thank the reviewer for this suggestion, we now modified the discussion of [11C]CH3 labeling in an appropriate order, and the discussion is consistent with what has been shown in now Figure 3B. This information can be found in the Results and Materials & Methods.

  1. Promethazine is a known drug with a high affinity for H1 receptors, some of then being located in brain areas, was this consider? could it be a drawback ? the point should be mentioned and somehow elucidated.

Answer: We thank the reviewer for this thoughtful notion. We showed the evidence of promethazine binding to Abeta in our previous study using a preclinical mouse model of Alzheimer’s disease (Neuroimage Clinical 2013, 2, 620-9), and this project is an extension of that work to validate the specificity of human hippocampal tissues. We hope that there will be no drawback regarding its affinity for H1 receptors. The primary and secondary targets have two distinct pathways; one is for therapeutic and the other focuses on probe development. Our job is to ensure a chance to repurpose this drug into a probe for Abeta imaging with no toxicity or interference with the primary target. And PET probe development is part of that solution since it can be administered at a low dose, under the threshold to initiate pharmacological responses.

  1. Figure 2 and 3 are inverted (wrong numbering, Figure 2 should appear before figure 3)

Answer: We swapped the order as suggested in this revised manuscript.

  1. Line 125 : this is not a novel chemistry (N-methylation using 11C is well known) , should be rephrased as "We have developed a synthesis of ......."

Answer: Yes, we agree with the reviewer that the chemistry for making a novel probe is more appropriate. This has been modified in the Discussion.

  1. The chemical reaction involving CH3I (should be written [11C]CH3I) is different from the data provided in the material and methods (100°C, 5 min on scheme and 90°C, 2 minutes in the material and method?)

Answer: We are sorry for this mismatch between the text and what has been shown in the figure. We have already revised it for consistency.

  1. For Figure 2B, as mentioned above, table should be in same order as the text.

Answer: We rewrite the results in the order as shown in, now figure 3B. And this newly added part can be found in the Results and Materials & Methods.

  1. The author indicate a specific activity at the end of synthesis of 48 TBq/mmol. the value could be added in the table, especially if a specific activity was also determined for the two other methods.

Answer: We thank the review for this suggestion. Yes, in the newly modified, now Figure 3B, we added the radiochemical yields and molar activity values.

  1. Figure 2C: is described in the bottom note as a preparative HPLC, while in material and methods it refers to an analytical HPLC co-injected with "cold" reference (line 253-260); Could you also please explain why in the preparative HPLC the UV signal reach a saturation (considering the low amount of precursor used this seems surprising).

Answer: We have confirmed for consistency that the data shown in, now Figure 3C is indeed analytical HPLC with co-injection of the “cold” standard. We do admit that there is a miscommunication between our team members.

  1. Line 154-156: Author state that use of NaH is ideal ....it is a common reagent to generate activated substrate " this is true only considering the acidity of the substrate (ie alcohols, thiols). Most 11C methylation of amines are carried out with weak base (NaOH, Na2CO3, CsCO3..). Moreover, the failure of the reaction with NaH and the success without the base demonstrate it. Why a reaction with a weak base was not attempted to compare NaH with another more suitable base (used only to trap HI)?

Answer: We are certain that using NaH is a common approach. We did not observe a good yield for this reaction condition. As shown in reference 23 (J. Org. Chem. 2009, 74, 2567-70), it provides very convincing evidence that might explain while the yield is low.

  1. line 163: Again, this sentence is very confusing: "Despite all these drawbacks...as a solvent". To what are you referring? the drawbacks are connected to a specific method (not commonly used, NaH) not to the general methylation process of secondary amines.

Answer: We removed that expression in the revised manuscript.

  1. No radiochemical yields or conversion are given, this is surprising, only product yield in mCi (official unit is Bq or MBq) is indicated in table 2B without knowing the initial activity introduced in the synthesis and if the results are based on [11C]CH3I or [11C]CH3The value should be indicated decay corrected or non decay corrected along with the total synthesis time (purification and formulation included).

Answer: We added the decay-corrected radiochemical yield in the now figure 3B.

  1. In material and methods: the process of methylation used by the authors should be clearly defined: general procedure is described referring to CH3I and CH3OTf while in the text the use of CH3OTf is not mentioned. What methylation agent was fused for each method? a specific activity of 48 TBq/mmol. was obtained, is it a single determination or an average value; further, 3.2.2 the solution method: this is a general procedure (base if used) with some difference between text and figure 2 (reaction time, temperature) and between HPLC description (analytical) and figure 2C (preparative).

Answer: We thank the reviewer for this notion to improve the clarity of the manuscript. This information is now modified with a clear description of each method of labeling; this can be found in the Results and Materials & Methods, as well as in now Figure 3B.

  1. Line 259 "further the ...using LC-MS/MS. this sentence is useless here and should be in the chemical characterization of the reference compound (with the observed value for MS peak) as for 11C-PMZ the MS value observed correspond only to the reference compound.

Answer: It is our oversight for that issue. It is now removed in the revised manuscript.

  1. in material and methods: Phenomenex C18(2), Phenomenex is a brand not a model of column, please indicate the name of the column used, to what refers the (2)? some more detail on HPLC would be welcome (HPLC model, injection loop volume, flow).

Answer: All of this information has been updated in the revised manuscript, and they can be found in the Materials & Methods.

  1. Retention time of 11C-PMZ should be indicated clearly in the material and methods section.

Answer: We added this information in the Materials & Methods as suggested.

  1. Other minor issues, Line 198: Na2SO4(Typo); Line 200-201: approximately = approximately (typo); line 201: dioxanes (why plurial?)

     Answer: We took care of all these issues.

  1. For Autoradiography: What was the specific activity of 11C-PMZ at the time of incubation?

Answer: The specific activity of [11C]PMZ was about 0.08 TBq/mg. We updated this information in the Materials & Methods.

Round 2

Reviewer 1 Report

There are multiple grammatical errors throughout the manuscript. An in depth proofreading should be done.

There is little to no reason given for the use of 2-butanone as a solvent for the loop method preparation. The authors should acknowledge that this is a  potential contributing factor in the low yields observed for this method. High boiling solvents, such as DMF, are a good alternative to 2-butanone.

The preparation of Compound 7 reports the use of HCl in Dioxane as the last step of the procedure. This would undoubtedly produce the hydrochloride salt product, but this is not mentioned nor reported in the name or characterization data. 

Author Response

  1. There are multiple grammatical errors throughout the manuscript. An in depth proofreading should be done.

Answer: We are sorry to see this since we have already done the proof-reading very carefully. We think it would be more straightforward if the reviewer points out where the grammatical errors are, and we can directly respond to these issues. 

There is little to no reason given for the use of 2-butanone as a solvent for the loop method preparation. The authors should acknowledge that this is a  potential contributing factor in the low yields observed for this method. High boiling solvents, such as DMF, are a good alternative to 2-butanone.

Answer: This question was not included in the critiques in the last review cycle. Therefore, we didn't have the chance to respond. Basically, the boiling point of 2-butanone is high enough for the labeling. When performing this kind of work, a number of other factors must be considered, including the stability of the compound, and thus very high boiling point solvents are not always the best solution. DMF could help, but again we are not sure until we have to try and prove it. It could be in our pipeline for the next experiment. 

3. The preparation of Compound 7 reports the use of HCl in Dioxane as the last step of the procedure. This would undoubtedly produce the hydrochloride salt product, but this is not mentioned nor reported in the name or characterization data. 

Answer: We confirm the precursor 7 is not an HCl salt.